# Heart Failure Therapies for the Prevention of HER2-Monoclonal Antibody-Mediated Cardiotoxicity: A Systematic Review and Meta-Analysis of Randomized Trials

**DOI:** 10.3390/cancers13215527

**Published:** 2021-11-03

**Authors:** Lauren J. Brown, Thomas Meredith, Jie Yu, Anushka Patel, Bruce Neal, Clare Arnott, Elgene Lim

**Affiliations:** 1Garvan Institute of Medical Research, Sydney, NSW 2010, Australia; e.lim@garvan.org.au; 2St Vincent’s Hospital, Sydney, NSW 2010, Australia; thomas.meredith@svha.org.au; 3University of New South Wales, Sydney, NSW 2031, Australia; jyu1@georgeinstitute.org.au (J.Y.); apatel@georgeinstitute.org (A.P.); bneal@georgeinstitute.org.au (B.N.); c.arnott@georgeinstitute.org.au (C.A.); 4Victor Chang Cardiac Research Institute, Sydney, NSW 2010, Australia; 5The George Institute for Global Health, University of New South Wales, Sydney, NSW 2050, Australia; 6Department of Cardiology, Peking University Third Hospital, Beijing 100191, China; 7Royal Prince Alfred Hospital, Camperdown, NSW 2050, Australia; 8School of Public Health, Imperial College London, London SW7 2BX, UK

**Keywords:** heart failure, prevention, breast cancer, HER2 therapy

## Abstract

**Simple Summary:**

Monoclonal antibodies targeting HER2 are used for the management of early and metastatic HER2-positive breast cancer. Approximately 10–15% of patients diagnosed with breast cancer will be HER2-positive. The incidence of heart failure in breast cancer patients is becoming increasingly problematic, owing to the ageing of the population and the growing number of cancer survivors. The aim of our review was to assess the published evidence for the use of cardio-prevention strategies in HER2-monoclonal antibody-mediated cardiotoxicity. Whilst in the assessed trials the use of heart failure therapies did not reduce the risk of trastuzumab-associated cardiotoxicity, there was a reduction in the mean change in LVEF and in the rates of interruptions to HER2 therapy in patients treated with beta-blockers. This highlights the possible applications for neurohormonal therapies to prevent cardiotoxicity and mitigate interruption to vital HER2-monoclonal antibody treatment.

**Abstract:**

Monoclonal antibodies including trastuzumab, pertuzumab, and antibody-drug conjugates, form the backbone of HER2-positive breast cancer therapy. Unfortunately, an important adverse effect of these agents is cardiotoxicity, occurring in approximately 10% of patients. There is increasing published data regarding prevention strategies for cardiotoxicity, though seldom used in clinical practice. We performed a systematic review and meta-analysis of randomized-controlled trials to evaluate pharmacotherapy for the prevention of monoclonal HER2-directed antibody-induced cardiotoxicity in patients with breast cancer. Online databases were queried from their inception until October 2021. Effects were determined by calculating risk ratios (RRs) and 95% confidence intervals (CI) or mean differences (MD) using random-effects models. We identified five eligible trials. In the three trials (*n* = 952) reporting data on the primary outcome of cardiotoxicity, there was no clear effect for patients assigned active treatment compared to control (RR = 0.90, 95% CI 0.63 to 1.29, *p* = 0.57). Effects were similar for ACE-I/ARB and beta-blockers (*p* homogeneity = 0.50). Active treatment reduced the risk of HER2 therapy interruptions (RR = 0.57, 95% CI 0.43 to 0.77, *p* < 0.001) with similar findings for ACE-I/ARB and beta-blockers (*p* homogeneity = 0.97). Prophylactic treatment with ACE-I/ARB or beta-blocker therapy may be of value for cardio-protection in patients with breast cancer prescribed monoclonal antibodies. Further, adequately powered randomized trials are required to define the role of routine prophylactic treatment in this patient group.

## 1. Introduction

In 2020, breast cancer was the most frequently diagnosed malignancy and the fifth leading cause of cancer-related death globally [1]. The human epidermal growth factor receptor 2 (HER2) protein is over-expressed due to HER2 gene amplification in 10–15% of breast cancers, leading to accelerated cell proliferation and poorer clinical outcomes prior to the routine use of HER2-directed therapies [2,3]. The development of the humanized murine monoclonal antibody targeting HER2, trastuzumab, in the early 1990s yielded a transformative therapy for this aggressive breast cancer subtype [4]. Since the adoption of this agent into the routine management of early stage HER2 positive breast cancer, adding trastuzumab to chemotherapy has led to a 37% relative improvement in overall survival and relevant patients can now expect a 10-year survival rate of up to 85%, as compared to 75% prior to trastuzumab use [5]. More recently, additional HER2 antibodies such as pertuzumab, and trastuzumab chemotherapy conjugates such as trastuzumab emtansine and trastuzumab deruxtecan, have been added to the treatment algorithm for late and early stage HER2 positive breast cancer, increasing the use of HER2 antibodies in this breast cancer subtype.

An adverse effect associated with the use of all HER2 antibody-based therapy is cardiac dysfunction, which occurs in 5–11% [6,7,8] of treated patients, with a higher incidence when two antibodies are used concurrently [9]. The manifestation of cardiotoxicity is a quantifiable reduction in the left ventricular ejection fraction (LVEF), with or without development of clinical heart failure. One of the mechanisms underlying the development of cardiotoxicity is the inhibition of the Neuregulin-1-activated pathway and ErbB2/ErbB4 heterodimerization, resulting in diminished pro-survival signalling by cardiomyocytes [10,11,12]. This is often compounded by the antecedent use of anthracyclines, one of the most commonly used chemotherapy drugs in breast cancer also associated with cardiotoxicity [13,14], with over a quarter of patients treated with both agents developing cardiotoxicity [15]. The observation that anthracycline-mediated cardiotoxicity can be predicted by the cumulative dose administered has led to restrictions on total life-time exposure [13]. In contrast, trastuzumab-mediated cardiotoxicity cannot be as reliably predicted. Retrospective analyses have demonstrated a correlation with age, history of coronary artery disease, hypertension, and frequency of trastuzumab administration [16]. Current guidelines are limited to the evaluation of LVEF 3-monthly whilst on trastuzumab therapy to monitor for signs of cardiotoxicity and guide treatment interruption [17].

Reduced left ventricular systolic function is independently associated with an adverse prognosis, and importantly, necessitates an interruption or cessation of HER2-directed antibody therapy, which precludes patients from achieving the full survival benefit offered by this intervention [7,18]. While shorter durations of adjuvant trastuzumab-based therapies have been trialed, the results are marginally inferior to the standard 1-year duration [6,7,8,19,20]. Consequently, the recommended 12-month duration of trastuzumab therapy for early stage HER2 positive breast cancer has remained unchanged for over two decades. In the advanced breast cancer setting where ongoing treatment is required, the vast majority require a HER2-directed therapy backbone, and needing to cease therapy due to cardiotoxicity is a devastating outcome. As such, there has been great interest in the potential utility of preventative heart failure therapies to mitigate the development of trastuzumab-associated cardiotoxicity and allow completion of guideline-directed trastuzumab therapy. Given the proven benefits in the treatment of heart failure with reduced ejection fraction (HFrEF) over recent decades, particular interest has focused on the potential role of beta-blockers, angiotensin-converting-enzyme (ACE) inhibitors, angiotensin-II-receptor blockers (ARBs) and mineralocorticoid receptor antagonists, and more recently angiotensin-receptor-neprilysin inhibitors (ARNIs) and sodium-glucose transporter 2 (SGLT2) inhibitors [21,22]. 

The identification of a safe and effective therapy to prevent trastuzumab-associated cardiotoxicity and mitigate interruption to an essential backbone therapy will be of great significance to patients with breast cancer. Use of primary prevention for trastuzumab-related cardiotoxicity is not commonly utilised. As such, we systematically reviewed the literature for research pertaining to the primary prevention of HER2-monoclonal antibody-mediated cardiotoxicity. Specifically, we sought to assess the efficacy and safety of oral heart failure therapies in reducing the incidence of HER2-monoclonal antibody mediated cardiotoxicity.

## 2. Materials and Methods

This systematic review was performed according to Preferred Reporting Items for Systematic reviews and Meta-Analyses (PRISMA) guidelines [23]. The systematic review protocol was prepared prior to the study and prospectively registered at the PROSPERO database (number CRD42021271951).

### 2.1. Eligibility Criteria

To be eligible for inclusion, studies needed to be randomized controlled trials (RCTs) comparing the use of oral heart failure agents to a placebo for the primary prevention of HER2-directed monoclonal antibody associated cardiotoxicity. Oral heart failure therapy included all agents consistent with current guideline recommendations for the treatment of HfrEF [24], namely beta-blocking agents, ACE-inhibitors, ARBs, mineralocorticoid receptor antagonists, ARNIs, and SGLT2 inhibitors. Studies needed to report cardiotoxicity outcomes or at least an objective quantification of left ventricular function such as the mean change in LVEF. The primary outcome of interest was cardiotoxicity, as defined by an absolute reduction in LVEF of ≥10% OR a reduction in LVEF to <50% from any baseline value [25]. Other outcome data of interest were change in LVEF, change in global longitudinal strain (GLS), interruption of HER2 therapy, heart failure hospitalizations, major adverse cardiac events, decline in renal function, change in cardiac specific biomarkers, and deaths.

### 2.2. Search Strategy

A systematic search of four databases (Embase, SCOPUS, Medline via PubMed and Cochrane Library) was conducted using the following combination of MeSH and keyword terms: breast cancer, trastuzumab*, pertuzumab, prevention, and cardiotoxicity. The wild-card search elicited by the use of the asterisk (*) allows for terms of trastuzumab-emtansine and trastuzumab-deruxtecan to be evaluated, and thus studies of antibody-drug conjugates of trastuzumab were encompassed by this search. The search terms utilised were chosen to allow for the broadest possible search into prevention strategies for cardiotoxicity in HER2-positive breast cancer, with the HER2-positive population being encompassed through the terminology of trastuzumab, pertuzumab, and breast cancer. Databases were queried from their inception until October 2021; included studies were restricted to the English language. The search strategy and results are detailed in Appendix A. To ensure comprehensive capture, an additional manual reference check of pertinent literature including recent review articles [26,27,28] was performed to identify additional studies. Unpublished data and ongoing trials were sought from the International Clinical Trials Registry Platform.

### 2.3. Data Collection

A standardized, pre-piloted form was used to extract data from included studies. Two reviewers (L.J.B. and T.M.) independently extracted data and then discussed discrepancies. Disagreement was settled through discussion with a third reviewer (C.A.).

### 2.4. Data Items

Extracted data included information pertaining to study type, methodology, population characteristics, interventions, and outcome measures. Pre-specified outcome data of interest included cardiotoxicity (as defined in Section 2.1 Eligibility Criteria), LVEF, GLS number of HER2 therapy interruptions, and adverse events.

### 2.5. Assessment of Risk of Bias in Included Studies

Studies were assessed for risk of bias and methodological quality using the Cochrane Collaboration’s RoB2 tool for assessing risk of bias in randomized controlled trials [29] and assigned a summary judgement of either ‘Low Risk,’ ‘High Risk,’ or ‘Some Concerns’. Intention-to-treat effects were analyzed for all outcomes relevant to this review.

### 2.6. Data Synthesis and Analysis

Data were analyzed using RevMan 5.4 software (The Nordic Cochrane Centre, Copenhagen, Denmark). Outcomes of interest included cardiotoxicity, as defined above, cardiac function as quantified by the left ventricular ejection fraction or global longitudinal strain, HER2 therapy interruption, and safety outcomes. Due to the paucity of studies and heterogeneity of outcome reporting, some results are presented qualitatively with relevant figures. To ensure homogeneity in data pooling, studies reporting outcomes consistent with our definition of cardiotoxicity, LVEF measured before and within 6 months of completing treatment, and HER2 therapy interruption, were deemed eligible.

Differences were expressed as relative risk (RR) with 95% confidence interval (CI) for dichotomous outcomes (incidence of cardiotoxicity, HER2 therapy interruptions and adverse events), and the mean difference (MD) with 95% CI for continuous outcomes (change in LVEF and GLS). The Mantel–Haenszel (M-H) random effects model was used for dichotomous outcomes and the inverse-variance random effects model was used for continuous outcomes. For pooled continuous outcomes, specifically LVEF, both change-from-baseline and follow-up values were analyzed for agreement. Where continuous data were reported as median and interquartile range, the mean and standard deviation were calculated according to the methods described by Luo et al. and Wan et al. [30,31]. Heterogeneity was assessed using the I^2^ statistic and by calculating the p-value for heterogeneity [32]. An I^2^ statistic of 0–25% was considered to reflect a low likelihood, 26–75% a moderate likelihood, and 76–100% a high likelihood of differences beyond chance. A *p*-value of <0.05 for heterogeneity was also deemed likely to reflect a high likelihood of differences beyond chance.

### 2.7. Certainty Assessment

Each article was evaluated for evidence quality using the GRADE system, which was used to guide assessment of the strength of recommendations pertaining to individual outcomes [33]. The summary of findings and certainty of the evidence by GRADE system are detailed in Appendix A.

## 3. Results

The systematic search yielded 2585 titles. Once duplicates were excluded, 90 reports were retrieved and after abstract and full-text analysis, 85 were excluded leaving five eligible studies [34,35,36,37,38]. The identification, screening, and inclusion process is illustrated in Figure 1.

A total of 85 reports were excluded following eligibility assessment. Of these, 70 reports were excluded as they did not meet the criteria for study inclusion; for example, they were single arm trials, published study protocols, or review papers. A total of six reports were excluded as they did not meet the intervention criteria; patients were either not treated with heart failure therapies in the prophylactic setting, treated with intravenous therapies, or were managed with non-pharmacologic strategies. Of the nine reports excluded as they did not meet the population criteria, these were studies of patients with HER2-negative breast cancer.

The risk of bias of the included studies were assessed using the RoB2 tool (Appendix A) [29]. Of the five included studies, three studies were deemed to be low risk of bias and two of the studies were classified as some concerns for bias. Study characteristics are reported in Table 1. All studies included utilized trastuzumab as the HER2-directed monoclonal antibody for management of HER2-positive breast cancer. No studies were identified that utilized pertuzumab or the antibody drug conjugates trastuzumab-emtansine and trastuzumab deruxtecan. We included three RCTs for the primary outcome (952 total patients) and a further two studies were included for evaluation of the secondary outcomes.

### 3.1. Baseline Characteristics

We identified five eligible trials; two studied the effects of an ACE-inhibitor, one the effects of an ARB, and four the effects of a beta-blocker. Comparators were placebo or no treatment. In the included studies, all participants were women and the mean age ranged from 46.5–57 years. One of the five studies included patients treated for metastatic cancer [34]. The other four studies only included women who were being treated with adjuvant trastuzumab for early breast cancer. In two studies, all women were treated with anthracyclines then sequential trastuzumab [36,38], two studies included some patients with anthracycline exposure [35,37] and one study did not report on anthracycline exposure. Mean follow-up ranged between 3–21 months.

The dose and duration of trastuzumab were variably reported by the studies. In the adjuvant cohorts in Guglin et al. and Boekhoet et al. and Sherafati et al., trastuzumab was administered for 52 weeks as per standard protocol, or until progression of disease in the metastatic cohort of Sherafati et al. Farahani et al. utilized an unusual loading dose trastuzumab at 4 mg/kg, followed by a maintenance infusion of 6 mg/kg for a total of 52 weeks. Pituskin et al. reported the average dose of trastuzumab administered over the course of the study, which was 102–109 mg/kg +/− 9 mg/kg over 52 weeks.

The doses of the study intervention are detailed in Table 1. For two studies, Guglin et al. and Boekhoet et al., the study intervention was administered on the same day of the first dose of the trastuzumab. In the Guglin et al. study it was continued for the duration of trastuzumab, being 52 weeks or until the end of trastuzumab therapy. For Boekhoet et al., the candesartan was continued until 26 weeks after completion of trastuzumab treatment. In the study of Pituskin et al., intervention was commenced within 7 days of the start of trastuzumab therapy and for the duration of trastuzumab treatment, stopped at cessation of trastuzumab. Sherafati et al. and Farahani et al. reported the use of study intervention commenced prior to trastuzumab and continued for 3 months.

### 3.2. Primary Outcome

Three studies comprising 952 patients reported data about the primary outcome of cardiotoxicity (Figure 2) with no effect on risk of this outcome for patients assigned active treatment compared to control (RR = 0.90, 95% CI 0.63 to 1.29, *p* = 0.57). Of these patients, anthracycline exposure occurred in 417 women (43.8%). For patients assigned to either an ACE-inhibitor or an ARB, there was no difference in the risk of cardiotoxicity compared with placebo (RR = 0.86, 95% CI 0.51 to 1.47, *p* = 0.59). Similarly, the use of beta-blocking agents did not significantly reduce the risk of cardiotoxicity compared with control (RR = 0.52, 95% CI 0.1 to 2.66, *p* = 0.43). Effects were similar for ACE/ARB and beta-blockade (*p* homogeneity = 0.50), however, there was a moderate level of heterogeneity in both individual ACE/ARB and beta-blockade subgroups (I^2^ = 48%, *p* = 0.14 and 64%, *p* = 0.1, respectively) and overall (I^2^ = 42%, *p* = 0.14).

### 3.3. Secondary Outcomes

#### 3.3.1. HER2 Therapy Interruption

Two studies comprising 740 patients reported on HER2 therapy interruption [35,37]. The reasons for interruption reported by Pituskin et al. were a decline in LVEF. In contrast, the reasons for interruption were not explicitly reported by Guglin et al. Active treatment reduced the risk of HER2 therapy interruption (RR = 0.57, 95% CI 0.43 to 0.77, *p* < 0.001) with similar findings for ACE-inhibitors and beta-blockade (*p* homogeneity = 0.97). The use of beta-blockers significantly reduced the risk of HER2 therapy interruption (RR = 0.54, 95% CI 0.36 to 0.83, *p* = 0.005). When analyzed separately, ACE-inhibitors or ARB did not significantly reduce drug interruption (RR = 0.55, 95% CI 0.29 to 1.04, *p* = 0.07) (Figure 3). There was low risk of heterogeneity influence across subgroups (beta-blockade I^2^ = 0%, *p* = 0.36; ACE-inhibitors I^2^ = 29%, *p* = 0.24; overall I^2^ = 0%, *p* = 0.5).

#### 3.3.2. Change in LVEF

All five studies comprising 1082 patients reported LVEF data (Figure 4). Of these patients, anthracycline exposure occurred in 488 women (45.1%), and for 65 women (6%) the status of anthracycline exposure was not reported. Analyses of both mean change in LVEF from baseline (Figure 4) and follow-up LVEF comparisons (Figure 5) were performed. Guglin et al. did not report follow up LVEF values, only change-from-baseline indices. Conversely, Boekhout et al. only reported follow up LVEF values without performing a change-from-baseline comparison. There was a significant difference in LVEF change-from-baseline favoring the use of both ACE-inhibitors/ARBs (MD −1.74%, 95% CI −2.18% to −1.3%, *p* < 0.001) and beta-blockers (MD −1.49%, 95% CI −2.82 to −0.16%, *p* = 0.03, *p* homogeneity = 0.73). There was a moderate level of heterogeneity overall (I^2^ = 56%, *p* = 0.05), predominantly driven by heterogeneity in the trials of beta-blocking agents (I^2^ = 56%, *p* = 0.08) with low heterogeneity in the ACE trials (I^2^ = 0%, *p* = 0.82).

Meta-analysis of studies that included follow-up LVEF values demonstrated an overall favorable effect of heart failure therapies, which were associated with a 2.24% greater LVEF than placebo (95% CI 0.53 to 3.94%, *p* = 0.01). When stratified by drug class, beta-blockade individually was not associated with a significant difference in follow up LVEF (MD 2.45%, 95% CI −0.45 to 5.35%, *p* = 0.10). ACE inhibitor/ARB agents demonstrated a borderline improvement in LVEF (MD 1.86%, 95% CI −0.02 to 3.75%, *p* = 0.05). Despite the absence of a significant subgroup effect (I^2^ = 0%, *p* = 0.74), overall heterogeneity was high (I^2^ = 71%, *p* = 0.01).

#### 3.3.3. Global Longitudinal Strain

One study of carvedilol (beta blocker) reported on GLS as a study outcome [38]. The use of carvedilol protected against deterioration in GLS. Specifically, trastuzumab therapy did not significantly influence GLS in participants receiving carvedilol (mean change from baseline −0.32 +/− 0.8, *p* = 0.88). However, participants randomized to control demonstrated a significant worsening of GLS (−17.17 +/− 1.5 before trastuzumab therapy to −16.35 +/−1.3, mean difference 0.82 +/− 0.8, *p* < 0.001). The mean change in GLS between arms (0.5%) was statistically significant *p* < 0.001.

#### 3.3.4. Serum Biomarkers

Two studies evaluated the serum biomarkers of troponin-T levels and brain natriuretic peptide (pro-BNP) for cardiotoxicity [36,37]. Whilst, Guglin et al. did not report the outcomes for these values in their study, the study conducted by Boekhoet et al. reported no association between the occurrence of cardiotoxicity and troponin-T or pro-BNP levels.

#### 3.3.5. Adverse Events

Adverse event outcomes were reported in three studies [35,36,37]. In general, the combination of trastuzumab together with heart failure therapies was well tolerated. Fatigue, hypotension, dizziness, headache, and cough were more common in participants assigned to heart failure therapy. One study reported a death in a patient receiving carvedilol due to unrelated neutropenic colitis, and in the same study [37], four additional deaths occurred from the progression of breast cancer.

Guglin et al. reported significantly more adverse events in patients allocated to ACE inhibitor (lisinopril) therapy compared to beta blockade or control. When compared to placebo, lisinopril was associated with fatigue in 26 vs. 16%, dizziness in 20 vs. 11%, headache in 8 vs. 3%, and cough in 11 vs. 4% (*p* < 0.05). Hypotension was noted in 13% of patients treated with lisinopril compared with 3% on placebo (*p* < 0.01). In this trial, the rates of adverse effects were similar in the carvedilol and placebo arms. Pituskin et al. reported increased incidence of renal dysfunction in patients receiving either ACE inhibitors (perindopril 16%) or beta blockers (bisoprolol 19%) compared to placebo (3%). In contrast to the study by Guglin et al., the rates of hypotension were similar across all arms. Boekhout et al. reported adverse events of Common Terminology Criteria for Adverse Events (CTCAE) grade 3 or higher. Grade 3 fatigue, dizziness, or hypotension were reported in less than 1% of patients in both active (ARB) and placebo arms. There were no reported CTCAE grade 5 adverse in the included studies.

#### 3.3.6. Other Outcomes of Interest

No studies reported specifically on heart failure hospitalizations or major adverse cardiac events.

## 4. Discussion

Several important observations can be drawn from our analysis, which included five randomized trials limited to beta-blockers, ACE-inhibitors, and ARBs. Although these oral heart failure therapies used specifically for primary prevention were not associated with a statistically significant reduction in the risk of HER2-directed antibody-associated cardiotoxicity as defined by each study, the use of these agents possibly prevents deterioration in left ventricular function as a continuous measure, and were associated with a significant reduction in HER2 therapy interruptions. Moreover, heart failure therapy is well tolerated when co-administered with trastuzumab, despite an increased incidence of mild adverse events, namely dizziness, fatigue, headache, and cough. No CTCAE grade 5 events were reported, and there was only one death in the active treatment arm, which was unrelated to therapy.

With respect to the primary outcome, a challenge to data-pooling was the heterogeneous definition of cardiotoxicity adopted by each study. For example, Boekhout et al. used ‘cardiac events’ as their primary outcome, which was defined as a decline in LVEF of >15% or to an absolute value <45%. Guglin et al. defined cardiotoxicity as either a decrease in LVEF of ≥10% in patients whose LVEF is ≥50%, or a decrease of ≥5% from baseline in patients whose LVEF decreases to <50%. Pituskin et al. defined cardiotoxicity as a reduction in LVEF by ≥10% or to <53%. Additionally, studies used different imaging modalities to quantify LVEF, though echocardiography was most utilised. Our definition of the primary outcome, being either a reduction in LVEF of ≥10% or a reduction in LVEF to <50%, is consistent with the international guidelines [25], and facilitates inclusion of studies with definitions of cardiotoxicity reflecting more severe left ventricular dysfunction. For example, a reduction from 56% (normal range) to 47% (mildly impaired left ventricular function), which reflects a clinically and prognostically significant change in LVEF, would not be captured by the definition used by Boekhout et al. As such, less sensitive definitions in prospective trials may lead to underestimation of the potentially protective effect afforded by heart failure therapy. Indeed, the observed prevention in LVEF deterioration afforded by heart failure therapy (mean difference ~2%) is well below the thresholds across all included definitions and may partly explain the absence of an observed benefit on cardiotoxicity as the clinical endpoint. It is also interesting to note that prophylactic heart failure therapy was associated with a clinically important reduction in HER2 therapy interruptions, suggesting that there is a subgroup of patients unable to proceed with therapy despite not fulfilling the criteria for cardiotoxicity, perhaps due to symptomatic heart failure with a sub-threshold decline in LVEF.

Although heart failure therapies were associated with a greater mean follow up LVEF and a smaller change from baseline, there was moderate to high unexplained heterogeneity, rendering the treatment effect estimate for each subgroup uncertain. It is also worth considering that echocardiographically-derived LVEF as a metric of ventricular function is influenced by multiple factors, such as acquisition quality and interobserver variability, which confound its reproducibility and sensitivity for quantification of subtle cardiac dysfunction in this population. GLS, which is semi-automatically calculated from echocardiographic images, accurately quantifies myocardial deformation with retrospective data, suggesting it is a more sensitive method of detecting early cardiac dysfunction in cancer patients [39]. GLS was only studied in one included trial, in which carvedilol therapy demonstrated a protective benefit against deterioration in GLS [38]. Recent results of a randomised study comparing GLS vs LVEF-guided institution of cardioprotective therapy demonstrated greater use of early cardioprotective therapy with the GLS-guided strategy, however, this did not result in a significant difference in objective LVEF between both arms at 1 year [40]. Further prospective assessment of this modality is needed.

Other biomarkers, including troponin T and brain natriuretic peptide for detection of early cardiotoxicity, have been reviewed in prospective trials [41,42,43,44]. To date, these have had varied results, with some trials demonstrating that they may not be useful as early predictors for cardiotoxicity in patients treated with trastuzumab. The data remain too limited for their routine use and larger trials with the use of these assessments is warranted.

Whilst the aim of our review was assessment of HER2-mediated cardiotoxicity, multiple previous anthracycline-associated cardiotoxicity prevention trials have also included sub-populations of patients treated with trastuzumab. For example, Elkhateeb et al. randomized 126 breast cancer patients treated with anthracycline chemotherapy, with or without trastuzumab, to either a combination of enalapril and carvedilol or placebo, and although the intervention group was associated with statistically fewer participants experiencing a reduction in LVEF ≥10% as assessed by cardiac MRI (1.9% vs. 12.5%, *p* = 0.04), only 24% of patients received trastuzumab and pre-specified subgroup analyses were not performed [45]. Additionally, the recently published Prevention of Cardiac Dysfunction During Adjuvant Breast Cancer Therapy (PRADA) trial [46], which assessed the use of ARB’s or beta-blockade to attenuate the decline in LVEF in patients receiving adjuvant anthracycline-containing regimens, only included 27 patients treated with adjuvant trastuzumab. Neither the use of candesartan nor metoprolol was associated with prevention of cardiotoxicity at 2 years. The SAFE trial (NCT2236806) is an ongoing phase III study comparing the effect of bisoprolol, ramipril, or both drugs compared to placebo on prevention of subclinical myocardial injury (early and late subclinical cardiotoxicity measured with traditional echocardiography) in breast cancer treated with (neo) adjuvant anthracyclines +/− trastuzumab. The most recent interim analysis, following the recruitment of 191 of 480 patients, led to the closure of the ramipril arm due to futility [47]. The results of this study, which will continue with only three arms, are awaited with great interest.

### 4.1. Limitations

The limitations of the included studies include the heterogeneous outcome definitions, incomplete outcome reporting, and variable participant exposure to anthracycline therapy. Within the included studies, the incidence of prior anthracycline exposure varied significantly; exposure rates were 23% in the study by Pituskin et al., 40.4% for Guglin et al., and 100% for Boekhout et al. The remaining two studies did not report anthracycline exposure status. Guglin et al. were the only group to report on exposure-stratified outcomes, and importantly demonstrated that although heart failure therapy did not result in a difference in cardiotoxicity across the entire cohort, there was a significant improvement in cardiotoxicity-free survival, protection against LVEF reduction, and treatment interruptions in the subgroup of patients with prior anthracycline exposure. The absence of stratified data in the remaining studies raises the possibility that this subgroup of patients with prior anthracycline exposure may experience benefit from prophylactic heart failure therapy, the observation of which is diluted when results are admixed with non-exposed participants. Our search was confined to studies investigating trastuzumab use, randomised control trials, and those in the English language. Thus, a limitation of the systematic review is the potential omission of other cardio-prevention trials in this space published in a different language or single-arm trials. Whilst this is a possible oversight of the design, randomized trials have the highest level of attributed evidence, and thus this is likely to be reflected in our article.

Nevertheless, there are several limitations of our meta-analysis. Firstly, the meta-analysis relies on published data from reported trials rather than on individual patients’ data. Secondly, some patient data were excluded from our meta-analysis as we were not able to extract the individual data of patients who received or did not receive anti-HER2 therapy from the trials. In addition, heterogeneity assessment is less reliable in a meta-analysis with less than 10 studies, although funnel plot asymmetry testing for publication bias did not suggest bias in our analysis. For this reason, sensitivity analyses exploring the robustness of the meta-analyses and meta-regression exploring study-level predictors were also not conducted.

### 4.2. Future Perspectives

The medical treatment for HFrEF has evolved significantly over the last several decades. Since one of the earliest landmark studies in 1986 demonstrated the mortality benefit associated with vasodilator therapy [48], multiple subsequent positive randomized studies of agents targeting different pathways in the maladaptive pathophysiological heart failure cycle have resulted in the additive multi-agent approach to therapy recommended by international guidelines today [24]. Specifically, this includes the use of sympathetic nervous system antagonists; renin-angiotensin loop inhibitors; and mineralocorticoid receptor antagonists and agents with vasodilatory, diuretic, chronotropic, and/or inotropic action [24]. Arguably the biggest developments in the last decade have been the introduction of neprilysin inhibitors [49] and SGLT2 inhibitors [50], both of which have demonstrated additive survival benefit over existing standard therapy. Many of these agents have been the subject of research for the primary prevention of anthracycline-associated cardiotoxicity, however, only dexrazoxane, an iron-chelating agent, and ACE-inhibitors, have demonstrated low- to moderate-quality supportive evidence [51]. Moreover, these data may not be applicable to patients receiving trastuzumab, who were only represented in small numbers in these trials, if at all.

Notably, none of the studies identified reported on the use of newer agents such as neprilysin or SGLT2 inhibitors. SGLT2 inhibitors may have promise, after demonstrating a profound reduction in cardiovascular death and heart failure hospitalizations in patients with heart failure with reduced ejection fraction [52,53]. These agents have several potential mechanisms through which they mediate cardioprotective effects, such as optimisation in ventricular loading conditions; improved cardiac energetics with augmentation in ketone availability; and reduction in cardiac injury, hypertrophy, and fibrosis [54]. Although most of the studies in our analysis included patients without pre-existing cardiovascular disease, perhaps refining the selection of patients for cardioprotective therapy based on baseline risk factors, such as pre-existing cardiovascular disease, hypertension and older age [55], will reduce the number needed to treat as a preventative approach. There is a need for further adequately powered prospective research into the role of RAAS blockage, beta blockers, and newer heart failure therapies in the prevention of trastuzumab cardiotoxicity, as well as a consistent approach to cardiotoxicity reporting and analysis.

## 5. Conclusions

In this systematic review of five randomized trials, the prophylactic use of RAAS blockade or beta blockers did not reduce the risk of trastuzumab-associated cardiotoxicity. These studies demonstrate the assessment of cardiotoxicity in patients through the modality of transthoracic echocardiography and LVEF assessment. Future cardioprotective trials of established and novel heart failure treatments are needed to definitively answer this question given the clinical need to prevent cardiotoxicity and mitigate interruption of trastuzumab therapy, which is a crucial treatment for patients with breast cancer. The evaluation of serum biomarkers and further imaging modalities to assess early-onset cardiotoxicity may also help to guide therapeutic approaches.

## Figures and Tables

**Figure 1 cancers-13-05527-f001:**
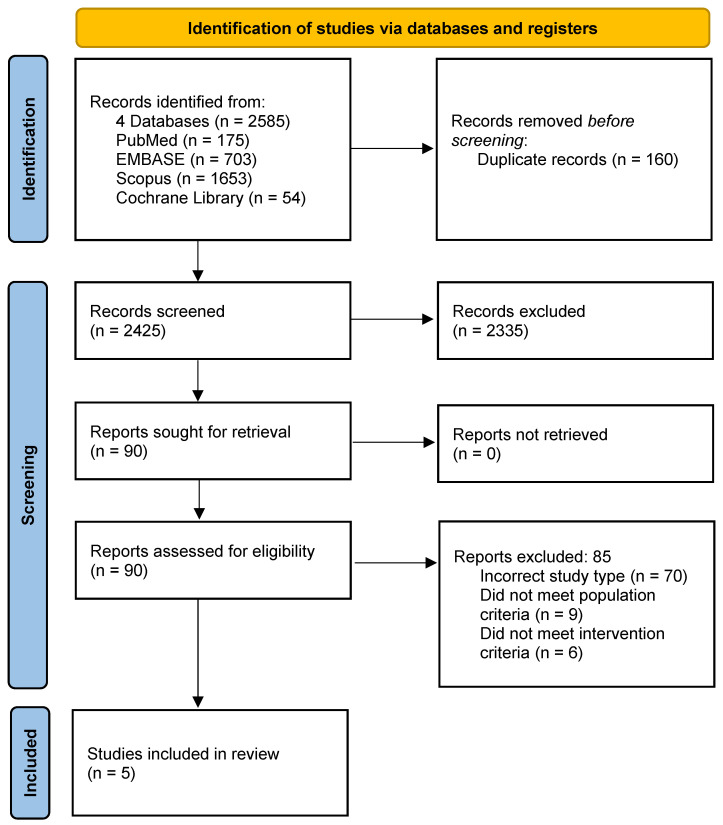
Systematic literature search flow diagram.

**Figure 2 cancers-13-05527-f002:**
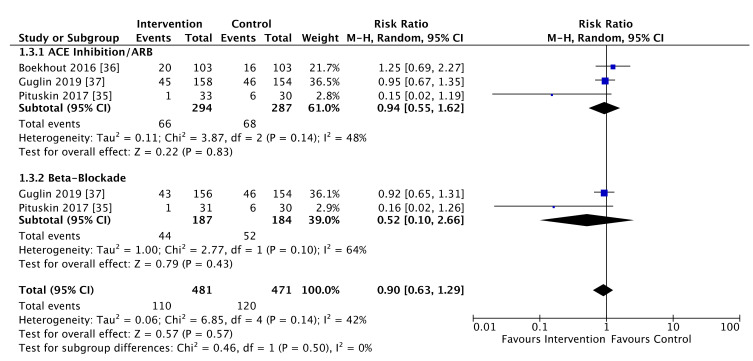
Forest plot illustrating pooled analysis of the primary outcome, cardiotoxicity, stratified by drug class.

**Figure 3 cancers-13-05527-f003:**
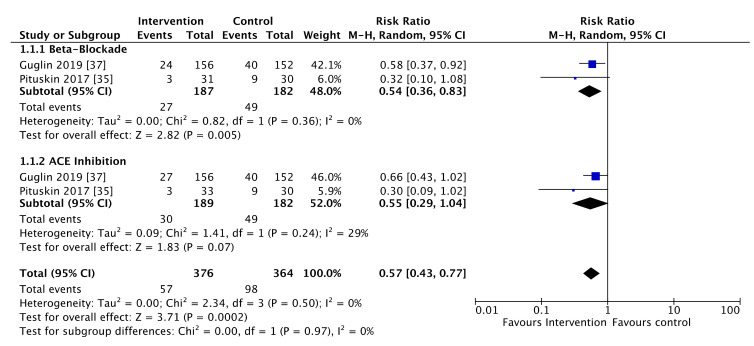
Forest plot illustrating pooled analysis of HER2 therapy treatment interruptions, stratified by drug class.

**Figure 4 cancers-13-05527-f004:**
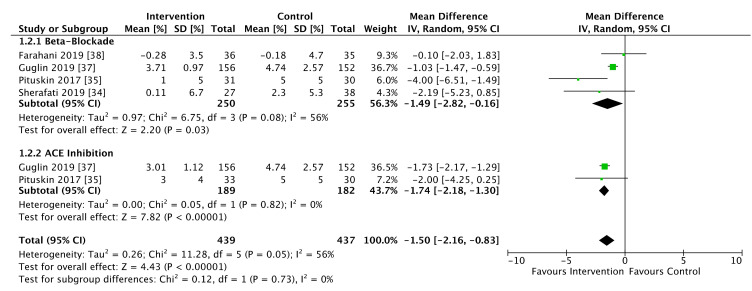
Forest plot illustrating pooled analysis of mean change in left ventricular ejection fraction (LVEF), stratified by drug class.

**Figure 5 cancers-13-05527-f005:**
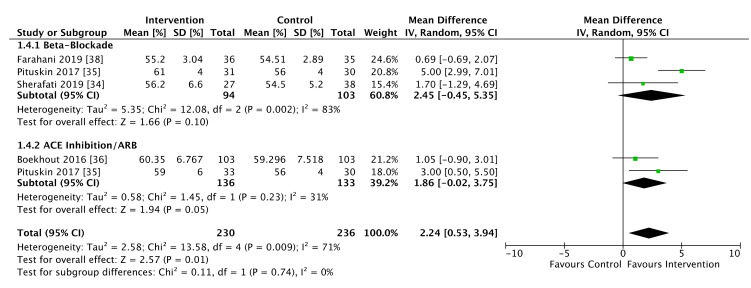
Forest plot illustrating pooled comparison of follow up LVEF, stratified by drug class.

**Table 1 cancers-13-05527-t001:** Characteristics of included studies.

Study	*n*	Study Design	Age, Mean +/− SD (Years)	Early or Metastatic	Anthracyclines	Type of Intervention	Intervention	Comparator	Follow Up	Outcomes
Boekhout et al. [36]	206	Double-blind RCT	49 +/− 24.1	Early	All	ARB	Candesartan 32 mg/day	Placebo	21 months	Occurrence of a cardiac event (decline in LVEF of >15% or an absolute value <45%) Safety of combination Biomarker evaluation
Farahani et al. [38]	71	Single-blind, open-label RCT	57 +/− 8.8	Early	All	Beta-blocker	Carvedilol 6.25 mg BD to MTD	No treatment	12 weeks	LVEF reduction GLS reduction LV diastolic function
Guglin et al. [37]	468	Double-blind RCT	51 +/− 10.93	Early	189 of 468 (40.1%)	ACE-inhibitor OR Beta-blocker	Lisinopril 10 mg daily OR Carvedilol 10 mg daily	Placebo	52 weeks	Rate of cardiotoxicity (LVEF decline of 10%) HER2 therapy interruption Treatment effects between anthracycline and non-anthracycline groups
Pituskin et al. [35]	94	Double-blind RCT	50 +/− 10	Early	22 of 94 (23.4%)	ACE-inhibitor OR Beta-blocker	Perindopril 2 mg daily OR bisoprolol 2.5 mg daily titrated to MTD	Placebo	52 weeks	LVEDVi Change in LVEF HER2 therapy interruption
Sherafati et al. [34]	65	Double-blind RCT	46.5	Early/Metastatic	NR	Beta-blocker	Carvedilol 6.25 mg BD	No treatment	12 weeks	Echocardiographic parameters

*n*, Number of participants; RCT, Randomized control trial; SD, Standard Deviation; LVEF, Left ventricular ejection fraction; GLS, global longitudinal strain; LVEDVi, LV end-diastolic volume index; SNP, single nucleotide polymorphisms; BD, twice daily; MTD, max tolerated dose.

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
