# Peer review of "Heart Failure Therapies for the Prevention of HER2-Monoclonal Antibody-Mediated Cardiotoxicity: A Systematic Review and Meta-Analysis of Randomized Trials"

_cancers, 2021, doi:10.3390/cancers13215527_

Round 1

Reviewer 1 Report

Dr. Brown et al. have performed a systematic review and meta-analysis of randomized-controlled trials to evaluate pharmacotherapy for the prevention of monoclonal HER2-directed antibody induced cardiotoxicity in patients with breast cancer. They conclude that prophylactic treatment may be of value for cardio-protection in patients with breast cancer treated by monoclonal antibodies.

Overall, the manuscript is interesting and well-organized. There are several minor comments

Minor comments:

  • What is the duration of prophylactic intervention in the studies? What is the dose and duration of trastuzumab treatment?

  • What is the indication of trastuzumab therapy interruption?

Reviewer 2 Report

This is a systematic review and meta-analysis of 35 randomized-controlled trials aimed to evaluate pharmacotherapy for the prevention of monoclonal HER2-directed antibody- 36 induced cardiotoxicity in patients with breast cancer.

The systematic review included five randomized trials, the authors conclude that the prophylactic use of RAAS blockade or beta blockers did not reduce the risk of trastuzumab-associated cardiotoxicity. Thus, further trials including novel heart failure treatments are needed.

The issue is relevant. The review is interesting and worthy, however there are some major criticisms, mainly methodological.

 Majors:

  1. Some relevant details listed in the PRISMA 2020 checklist are lacking. I recommend following the Checklist items and reporting the missing ones. Furthermore, It will beneficial to use the paragraph order (and names) defined by the PRISMA 2020 checklist.
  2. I would recommend providing an explicit statement of the review aims, as stated by PRISMA 2020 Guidelines. For instance: the aim was to assess the efficacy of pharmacological therapy for heart failure to prevent HER2-monoclonal antibody mediated cardiotoxicity in patients with breast cancer.
  3. Line 133-136: The Authors should describe how and why they chose the keywords for the literature searches. How do they ensure to include all the synonyms and variants of the keywords? Is HER2-directed monoclonal antibody included? I would recommend adding the full search strategies for all databases in the supplementary materials.
  4. Figure 1: the Authors excluded nine reports because they did not meet the population criteria. Please explicit and describe such unmet criteria.
  5. Lines 156-161: it will be beneficial to distinguish between the risk of bias and the level of evidence assessments. The risk of bias assessment should be rated and presented before the evaluation of the quality of the evidence.
  6. Line 156-160: To improve the readability, the authors should present the possible CEBM levels and methods used to define them. Then, I do not see the findings of CEBM level of evidence in the results section.
  7. Lines 170: The Authors used the Mantel-Haenszel (M-H) random effects model only. However, this method could be used with the dichotomous outcomes only. The authors adopted the inverse-variance random effects model for continuous outcomes, as shown in Figure 4. This information has to be added in the “Data analysis” paragraph.
  8. The included studies (TABLE 1) have a high heterogeneity, in terms of therapies, follow-up intervals, outcomes, as well as how the outcomes are presented (follow-up values, absolute changes from baseline, relative change from baseline). The Authors should describe in details the criteria used to define sufficiently homogenous data for meta-analysis.
  9. The Cochrane guidelines recommend using follow-up values, instead of changes from baseline, to perform meta-analysis of continuous outcomes, as LVEF (Figure 4). The Authors should justify their choice of using mean changes. To verify the consistency of their findings, the Authors could show the meta-analysis findings with follow-up values.
  10. Boekhout et al. (2016) reported in the supplementary materials baseline and follow-up LVEF values, that have not been considered in the review. The Authors should justify their choice and comment the LVEF findings of Boekhout and collegueas.
  11. Line 258: The Authors state that “Three studies reported adverse events”. However, one of these three studies (Pituskin et al 2017) found no significant drug related adverse effects.
  12. In the Discussion, the Authors should describe the main limitations of the present systematic review keeping those separated from the limitations of the included studies.
  13. In the conclusion, it is worth commenting about the use of LVEF assessed by echocardiography as the main parameter to quantify the occurrence of heart failure/LV dysfunction. For instance, the identification of biomarkers would greatly help guide therapeutic approaches.

 Minors:

  1. Line 49: the chosen keywords are very generic and misleading
  2. Line 104: “The identification of a safe, effective, and therapy”: something is missing after and..
  3. Line 139: I suggest adding references of all pertinent review articles checked.
  4. Line 119: what do the Authors mean for “referent control”?
  5. Lines 125-130: I would suggest describing the included outcomes in a paragraph called “Data items”, instead of “Study selection”.
  6. Line 135: It is unclear why the symbol * was used for “trastuzumab”.
  7. Lines 157-160: I suggest adding the criteria used to assign the summary risk of bias judgement in paragraph 2.4 or in the Supplementary Material Table 1, in order to justify the results reported in lines 182-184.
  8. Line 164-175: The authors should indicate for each outcome and class of drug the effect measure (risk ratio, mean difference, etc.) they used, as recommended by PRISMA guidelines.
  9. Line 117-130: Did the authors exclude HER2-negative breast cancer? In CECCY trial this was included however it is not mentioned in the present review.
  10. Figure 1: there is a strong suggestion including the number of records identified from registers and citation searching, as shown in PRISMA 2020 flowchart. It allows finding the link between the methods described paragraph 2.2 and the results in Figure 1.
  11. Figure 1: boxes and arrows of the flow diagram are missing.
  12. Table 1: “BD” and “MTD” should be explained in the legend.
  13. Table 1: some important information are missing, the suggestion is to rearrange the table in the following order: study design (for example: single- or blind-, cross-over, parallel group RCT), participants characteristics, type of intervention/comparator and outcomes, also adding other relevant data if the case.
  14. Table 1: The Authors should specify the class of drugs (i.e. ACE-inhibitors/ARB or beta-blockade) used in the included studies.
  15. Table 1: mean age should be completed with the standard deviation.
  16. Please use the terms referred to the studies homogeneously throughout the manuscript, in figures, tables, and supplementary materials. Adding reference numbers after the names of the studies in the forest plots would help.
  17. Lines 182-184: The Authors state that the overall ratings of risk of bias is shown in Figure 2. I do not see them in Figure 2.
  18. Figure 3: The Authors should check the sample size of the control group in the study of Sherafati et al. 2019 [29]. The Authors reported n=39, instead of n=38.

Round 2

Reviewer 2 Report

Major criticisms have been solved. The manuscript has been improved.

There are still some minors to be corrected:

  • Line: 31-48: The Authors should check and update the abstract. They should change the date of the last literature search (line 38), explicit the name of the primary outcome (line 40) and use the “Mean Difference” instead of “weighted mean difference” (line 39). 

  • Figure 5: To improve readability, the Authors should invert ‘Favours Control’ and ‘Favours Intervention’ in the forest plot. In the previous figure 1-4, ‘Favours Control’ was always on the right side of the forest plot, this rule should be used in Figure 5 too. 

  • Line: 180-182: The revised version of the manuscript includes a new paragraph, called serum biomarkers. The Authors should include serum biomarkers among the pre-specified outcomes.  
  • In Lines 180-182, 206-208, 196-198, in Paragraph names 3.3.1-3.3.5 and figure legends, the authors used different names to refer to the same outcome measure. The Authors shoud use a unique name for each included outcome to improve readability.   
